**Data Availability Statement:** All relevant data are within the paper and its Supporting Information files.

# An assessment of the impacts of litter treatments on the litter quality and broiler performance: A systematic review and meta-analysis

**Taiani dos Santos de Toledo[1], Aline Arassiana Piccini Roll[1], Fernando Rutz[1], Henrique Müller Dallmann[2], Marcos Antonio Dai Prá[3], Fábio Pereira Leivas Leite[4], Victor Fernando Büttow Roll** [1] *

1 Department of Animal Science, Faculty of Agronomy Eliseu Maciel, Federal University of Pelotas, Pelotas, Brazil, 2 Ibirubá Campus, Federal Institute of Education, Science and Technology of Rio Grande do Sul, Ibirubá, Brazil, 3 Brasil Foods S/A. Av. Presidente Vargas, Marau, Brazil, 4 CDTec, Biotechnology, Federal University of Pelotas, Pelotas, Brazil

* roll98@ufpel.edu.br

## Abstract

### Objective

The choice of the most suitable litter treatment should be based on scientific evidence. This systematic review assessed the effectiveness of litter treatments on ammonia concentration, pH, moisture and pathogenic microbiota of the litter and their effects on body weight, feed intake, feed conversion and mortality of broilers.

### Methods

The systematic literature search was conducted using PubMed (Medline), Google Scholar, ScienceDirect and Scielo databases to retrieve articles published from January 1998 to august 2019. Means, standard deviations and sample sizes were extracted from each study. The response variables were analyzed using the mean difference (MD) or standardized mean difference (SMD), (litter treatment minus control group). All variables were analyzed using random effects meta-analyses.

### Results

Subgroup meta-analysis revealed that acidifiers reduce pH (P<0.001), moisture (P = 0.002) ammonia (P = 0.011) and pathogenic microbiota (P <0.001) of the litter and improves the weight gain (P = 0.019) and decreases the mortality rate of broilers (P<0.001) when compared with controls. Gypsum had a positive effect on ammonia reduction (P = 0.012) and improved feed conversion (P = 0.023). Alkalizing agents raise the pH (P = 0.035), worsen feed conversion (P<0.001), increase the mortality rate (P <0.001), decrease the moisture content (P<0.001) and reduce the pathogenic microbiota of the litter (P<0.001) once compared to controls. Superphosphate and adsorbents reduce, respectively, pH (P<0.001) and moisture (P = 0.007) of the litter compared to control groups.

**Funding:** The Brasil Foods S/A provided financial support in the form of author' salary to MADP but did not have any additional role in the study design, data collection and analysis, decision to publish, or preparation of the manuscript. The authors Fábio Leivas Leite and Victor Fernando Büttow Roll were supported by grants from Conselho Nacional de Desenvolvimento Científico e Tecnológico, Brazil (CNPq/Produtividade em Pesquisa). The Author Aline Arassiana Piccini Roll was supported by grant from CAPES - Coordenação de Aperfeiçoamento de Pessoal de Nível Superior, Brazil. The funders had no role in study design, data collection and analysis, decision to publish, or preparation of the manuscript.

**Competing interests:** The MADP received salary from Brasil Foods S/A. This does not alter our adherence to PLOS ONE policies on sharing data and materials. The other authors declare no potential conflict of interest.

## Conclusion

None of the litter treatments influenced the feed intake of broilers. Meta-analyses of the selected studies showed positive and significant effects of the litter treatments on broiler performance and litter quality when compared with controls. Alkalizing was associated with worse feed conversion and high mortality of broilers.

## Introduction

The litter must provide comfort and well-being so that broilers can express their full genetic potential as they remain housed on it for most of their life. In this context, the litter must be treated properly to control proliferation of insects, growth of pathogenic microorganisms, moisture and the production and volatilization of ammonia [1].

Litter reuse during several consecutive flocks is a management practice that has been widely adopted in the production of broilers. Reusing the litter reduces production costs, minimizes the problem of material availability and decreases the amount of waste generated by the production of chickens [2], in addition to maintaining or even improving the performance of animals. However, it is necessary to adopt efficient litter treatments to reduce risks to human and poultry health [1]. Acidifiers, alkalizers, adsorbents, agricultural gypsum and superphosphate are the conditioners most used to treat poultry litter. The conditioner chosen must be able to reduce negative points and enhance the favorable characteristics of the poultry litter.

However, there are many divergences among the results found in the scientific literature on the effects of litter treatments on broiler production performance and litter quality. The objective of this study was to determine the comparative effectiveness of the litter treatments for improve litter quality and broiler performance.

## Materials and methods

In this review, the bibliographic search was carried out with broad search criteria, to reduce the number of false negatives as much as possible (important studies not found in the search phase), while increasing the number of false positives (studies found during the search that do not meet the inclusion criteria) [3].

The meta-analysis methods presented in this study have already been described and published by other authors [4, 5, 6, 7]

### Literature search strategy

The literature search for this study was carried out between June and August 2019 using the following electronic databases: PubMed (Medline), Google Scholar, ScienceDirect and Scielo. The searches were run by combining the keywords: "Poultry litter treatment" and "Broilers"; "Poultry litter treatment" and "Chickens"; "Poultry litter amendment" and "Broilers"; "Poultry litter amendment" and "Chickens"; in studies published in English, Spanish or Portuguese.

Two researchers independently performed the search and selected the studies according to the pre-established inclusion and exclusion criteria. Although the search was extensive, authors were not contacted to ascertain further information or to obtain unpublished work. However, the possibility of publication bias due to the existence of unpublished studies was evaluated using the funnel plot technique and Egger's test.

The studies were selected using a two-step approach, first by analyzing the title and abstract then analyzing the full-text.

The extraction of quantitative data for the meta-analysis of the included studies were compared and selected in agreement with the researchers. If there was no consensus, a third researcher was required to answer any discrepancy by further, in-depth analysis.

### Study eligibility criteria

To be included in the systematic review, studies should meet the following criteria: full articles from peer-reviewed journals published between 1998 and 2019; evaluation of different litter treatments compared to control groups (no treatment); sufficient quantitative data to calculate the effect size and a complete description of the experimental design were required. The following were excluded from the systematic review: studies published prior to 1998; studies with laying hens, broiler breeders or carried out *in vitro* or in the laboratory without the presence of animals; studies which did not present averages and measures of variability, studies in which litters or birds were experimentally inoculated with diseases.

### Extraction of quantitative data

To reduce heterogeneity between studies when results from several flocks were presented (litter reuse) in the same article, values were always extracted from the newest flock (varying from the 1st to the 5th flock).

The response variables that were extracted for the meta-analysis included: weight gain, feed intake, feed conversion and mortality rate of broilers, while for litter quality characteristics included concentration and volatility of ammonia, pH, moisture and existing pathogenic microbiota.

Additional data were extracted, such as: features of the published study (author, year and place of publication), characteristics of the animals (sample size, age, breed and stocking density (birds/m$^2$), characteristics of treatments (conditioner and application dose) and number of litter reuses.

### Statistical analysis

A Der-Simonian-Laird random effect model [8, 9] was used to analyze the extracted data. The random effects model considers the existence not only of the variation within each study, but also the variation between the studies, that is, it considers that the effects of the studies are not the same, but that they are connected through a distribution of probability [10, 11, 12]. Each article included in the meta-analysis was considered as a random sample belonging to a larger population of articles [4, 5]

From the articles selected for the meta-analysis, the mean, standard deviation and number of replicates in each treatment were extracted. When the articles presented the standard deviation (SD) for each group, these values were used directly in the meta-analysis. When a single common measure of accuracy was presented for all group means, the same value was used for the treated and control groups in the meta-analysis. In cases where standard deviation was not reported, it was calculated from the standard error of the mean, coefficient of variation or confidence interval.

Due to the distinct nature of the response variables analyzed, the calculation was performed in two ways: through the mean difference (MD) between the treated group and control and by the standardized mean difference (SMD). The SMD analysis was computed through the differences between each of the treated litters and the untreated divided by the pooled standard deviation.

SMD = Treated litter–control /pooled standard deviation

A MD or SMD of zero indicated that the treated litter and the control had equivalent effects.

**Subgroup analyses.** Subgroup analyses were carried out by splitting different litter conditioners into subgroups according to their characteristics in order to investigating heterogeneous results and making comparisons between these groups. The acidifiers were formed by aluminum sulfate, sodium bisulfate, potassium permanganate, aluminum chloride, ferrous sulfate, acidified clay, alum, hydrochloric-citric phosphoric acid and SoftAcid™.

The alkalizing sub-group was formed by hydrated lime, quicklime, calcitic limestone and dolomitic limestone. Sepiolite, zeolite, bentonite and coal belonged to the subgroup of adsorbents. In turn, gypsum and superphosphate, as they do not fit the characteristics of the others, were analyzed in separate subgroups.

It was identified that for some treatments and response variables there was only a single published study. For this reason, superphosphate was excluded of subgroup meta-analysis of weight gain, feed intake and feed conversion [13] and moisture [14].

The fermentation subgroup was excluded from the meta-analysis, because [15], it was the only work for the variables weight gain, feed consumption and feed conversion, [16] the only study for ammonia, pH and moisture and [17] the only publication for pathogenic microbiota.

**Forest plots.** The effects of treatment on poultry litter were presented in forest graphics. For weight gain, feed intake, pathogenic microbiota and concentration and volatility of ammonia the SMD for random effects model was used. For feed conversion, mortality, pH and moisture, the MD for random model effects was used.

The vertical line indicates zero difference or no effects (between the treated group and the untreated group). Points to the left of the solid vertical line represent a reduction of response variable, while points to the right of the line indicate an increase of response variable on treated litter. The total size of the combined effect and the 95% confidence interval are indicated by the diamond at the bottom of the forest plot.

**Analysis of heterogeneity.** The measurement of heterogeneity was performed using the Chi-square test (Q) and the $I^2$ statistic [18]. Values of $I^2$ range from 0% to 100% and values close to 25% indicate low heterogeneity, close to 50% indicates moderate heterogeneity and close to 75% indicates high heterogeneity between studies [10, 11, 19, 20]. Values greater than 50% indicated significant heterogeneity [5].

**Publication bias.** Publication bias was assessed applying Egger's test [21] and funnel plots using the meta package [22]. The Egger test quantifies the funnel plot asymmetry and performs a statistical test. The funnel plot shows the presence of possible publication bias which could be attributed to studies with small sample size and with negative results that were not published [23].

## Results

### Literature search and extracted information

The literature searches in the electronic databases for studies published in the last 21 years (1998–2019) resulted in 5891 reports. After removing duplicate publications, the total was 5866 articles. A selection of these articles was made in relation to the title and abstract, which resulted in 5801 excluded articles. The remaining 65 publications were evaluated according to the full text. Of these, 35 articles met the pre-established inclusion criteria. Thus, after removing articles that did not present variability measures, or that did not have repetitions for subgroups, or that used a health challenge in birds, 26 articles remained for the meta-analysis.

In Fig 1, it is possible to identify the number of articles selected and excluded at each phase of the systematic review and meta-analysis.

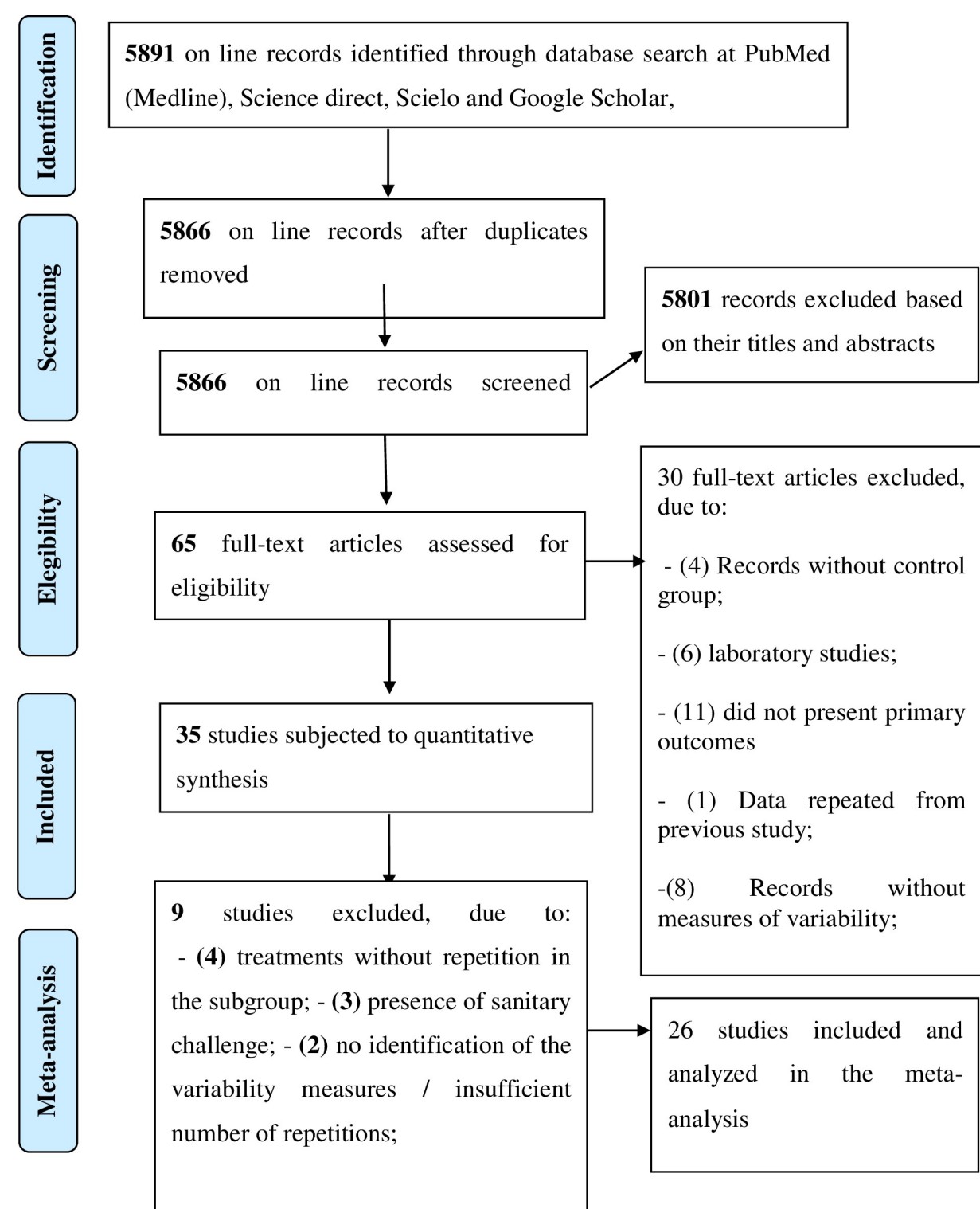

**Fig 1. Diagram adapted from the PRISMA-P guidelines [24], showing the total number of reports identified and the number of reports filtered at each stage of the study selection process from the systematic review on poultry litter treatments.**

Three articles submitted the animals to a sanitary challenge, before evaluating the conditioner used in the treatment of the litter. Thus, [25], [26] and [27], were excluded from the meta-analysis for challenging animals with *Escherichia coli*, coccidiosis and ammonia, respectively.

The reference [28] was excluded from the meta-analysis because the variability measures were not shown. The study [29] was excluded because presented only one repetition for both the treated group and the untreated group.

Some litter treatments did not have more than one published article so they were excluded from the meta-analysis, namely: enzymatic biocatalyst (Rydall) and bacterial culture (Mizuho) [30]; *Yucca Shidigera* extract [31]; non-absorbent polymers [32] and biodegradable treatment [33].

Table 1 Shows the characteristics of the studies included in the meta-analysis.

## Feed intake

The forest plot (Fig 2) showed that there was no statistical difference between the treated group and the untreated group for feed intake (SMD = -0.003, IC = -0.206–0.199, P = 0.973).

Thus, litter treatments did not affect the feeding behavior of the birds. For the overall effect of litter treatments on feed intake, non-significant heterogeneity ($I^2$ = 0% and P = 0.576) was observed. Therefore, the analysis of subgroups also shows that there is no effect of litter treatment on feed intake (P> 0.05).

## Weight gain

Weight gain in broilers reared on treated litter is shown in Fig 3. The diamond in the forest plot shows an overall positive effect on the weight gain of broilers reared on treated litter compared to untreated litter (SMD = 0.366, IC = 0.027–0.705, P = 0.034). This effect was heterogeneous, as indicated by $I^2$ = 65.76% and P value <0.001. Subgroup analysis revealed a favorable response to weight gain in broilers reared in litters treated with acidifiers (SMD = 0.570, IC = 0.095–1.044, P = 0.019).

The litters treated with adsorbents, alkalizers and gypsum did not differ statistically from the control group (P> 0.05). However, in the forest plot, it is possible to see a positive tendency for weight gain in birds reared on litters treated with adsorbents and gypsum, and a negative tendency in those reared on litters treated with alkalizers when compared to untreated litters.

## Feed conversion

In Fig 4, it is possible to identify that the use of conditioners in litter compared to untreated litter did not show any statistical difference, although it is possible to see a positive trend for improvement in feed conversion in the treated group (MD = -0.013, IC = -0.029–0.002, P = 0.086).

This effect was heterogeneous, as indicated by $I^2$ = 78.68% and P <0.001. In the analysis of subgroups, birds raised in litter treated with gypsum (MD = -0.063; IC = -0.117 - -0.009, P = 0.023) showed better feed conversion than those in the control group. In contrast, alkalizers (MD = 0.034, IC = 0.015–0.053, P <0.001) had a worse feed conversion.

The feed conversion of birds reared on litter treated with adsorbents was similar to those reared on untreated litter (P> 0.05). Although acidifiers did not differ statistically from the control group (MD = -0.018, IC = -0.037–0.001, P = 0.067) they showed a tendency to improve feed conversion.

**Table 1. Details of the 26 studies included in the meta-analysis.**

| Author (year) | Country | Conditioners | Dosage | Poultry litter | Breed | Birds/ $m^2$ | Response variables |
|---|---|---|---|---|---|---|---|
| [34] Avcilar et al. (2018) | Turkey | SEP | 25 and 50% | 2 and 1 | Ross | UN | WG, FI, FC, $NH_3$ $NH_3$, pH, Mo |
| [14] Bordignon (2013) | Brazil | G, LI, SP, AS, HL | 0.500, 0.500, 0.500, 1.000 e 0.300 kg | 1 | UN | 16 | $NH_3$, pH, Mo |
| [35] Bruno et al. (1999) | Brazil | G | 5, 10, 15 and 20 kg | 1 | Hubbard | 9 | WG, FI, FC, Mort |
| [36] Celen and Alkis (2009) | Turkey | ALU | 0.091 kg/Bird | 3 and 6 | Ross | UN | pH |
| [37] Chung et al. (2015) | South Korea | AS | 100g/kg | 2 | Arbor Acres | UN | $NH_3$, pH, Micro |
| [38] Do et al. (2005) | South Korea | FS, AS, AC, Alum + $CaCO_3$, AC+$CaCO_3$, PP | 1.150, 1.150, 1.150, 1.150, 1.150 and 0.115 kg/$m^2$ | 4 | Arbor acres | UN | pH, Mo |
| [13] Ferreira et al. (2004) | Brazil | AS, G, SP, HL | 100 g/kg, 40% of total weight, 30 kg/to, 0.5 kg/$m^2$ | 1 | UN | 12 | WG, FI, FC |
| [39] Furlan (2017) | Brazil | AS | 200, 400 and 600g/$m^2$- 200 400 and 600g/$m^2$ -1568g/$m^2$ | 2 | Cobb | 8–9–8 | WG, FI, FC, $NH_3$, pH |
| [40] Garrido et al. (2004) | Norway | SAC | 7% of bird weight | 1 | Ross | UN | FI, FC, pH, Mo, Micro |
| [41] Li et al. (2013) | United States of America | SB | 244g/2 weeks/$m^2$ | 1 | Ross | UN | FC, Mort, $NH_3$, pH, Mo |
| [16] Loch et al. (2011) | Brazil | COMP, AS, G, QUI, DL, ZEO, CH | -, 0.56kg/$m^2$, 40% of total weight, 0.5kg/$m^2$, 1.5kg/$m^2$, 5% of total weight, 20% of total weight | 5 | Ross | 10 | $NH_3$, pH, Mo |
| [17] Lopes et al. (2015) | Brazil | QUI, TAR, QUI+TAR | 300g/$m^2$, —, 300g/$m^2$ | 1 | Cobb | 12 | Micro |
| [42] Madrid et al. (2012) | Spain | AS | 0.25kg/$m^2$ | 1 | Ross | 9 | $NH_3$ |
| [43] McWard and Taylor (2000) | United States of America | ACC, ALU, SB ALU, ACC, ACC | UN | UN | Cobb X Hubbard Cobb x Cobb | UN | WG, FC |
| [44] Nagaraj et al. (2007) | United States of America | SB | 0.02kg/$m^2$ in day 1, 0.04kg/$m^2$ in day 1, 0.02kg/$m^2$ in day 1 in day 21 | 1 | UN | 10 | FC, Mort, Mo |
| [45] Oliveira et al. (2003) | Brazil | AS, G, HL | 490g/$m^2$, 40% of total weight, 0.5kg/$m^2$ | 2 | UN | 12 | $NH_3$, pH |
| [46] Oliveira et al. (2004) | Brazil | AS, G, SP, HL | 100g/kg, 40% of total weight, 30 kg/ton, 0.5kg/$m^2$ | 1 | UN | 12 | $NH_3$, pH |
| [15] Oliveira et al. (2015) | Brazil | COMP, AS, G, QUI, DL, ZEO, CH | -, 0.56kg/$m^2$, 40% of total weight, 0.5kg/$m^2$, 1.5kg/$m^2$, 5% of total weight, 20% of total weight | 5 | UN | 10 | WG, FI, FC |
| [47] Purswell et al. (2013) | United States of America | SB | 0.48kg/$m^2$ in day 1, 0.48kg/$m^2$ in day 1 and 28, 0.48kg/$m^2$ in day 1, 14, 28 and 43 e 0.48kg/$m^2$ in day 1, 24 and 43 | 1 | UN | UN | WG, FI, FC, Mort, $NH_3$ |
| [48] Ruiz et al. (2008) | United States of America | QUI | 10 e 15% | 1 | UN | 12 | FI, FC, Mort, pH, Mo |
| [49] Sahoo et al. (2017) | India | ALU, SB | 90 and 25g | UN | Vencobb | UN | WG, FI, FC, pH, Mo, Micro |
| [50] Sampaio et al. (1999) | Brazil | G | 5, 10, 15 and 20kg | 1 | Cobb | 9 | $NH_3$, Micro |
| [51] Taherparvar et al. (2016) | Iran | BEN, QUI | 3 and 1.5kg/$m^3$ | UN | Ross | UN | WG, FI, FC, pH, Mo, Micro |
| [52] Tasistro et al. (2007) | United States of America | AFC, SB | 1:25, 0.244kg/$m^2$ | 6 and 1 | Cobb | UN | pH |

*(Continued)*

**Table 1.** (Continued)

| Author (year) | Country | Conditioners | Dosage | Poultry litter | Breed | Birds/ m² | Response variables |
|---|---|---|---|---|---|---|---|
| [53] Toppel *et al.* (2019) | Alemanha | SB | 250 and 150g/m² | UN | Ross | UN | pH |
| [54] Zhang *et al.* (2011) | China | ALU | 1 kg/m² | 2 | Arbor Acres | 12–16–20 | WG, FI, FC, pH, Mo |

SEP = sepiolite; G = gypsum; LI = limestone; SP = superphosphate; AS = aluminum sulfate; HL = hydrated lime; ALU = alum; FS = ferrous sulfate; AC = aluminum chloride; Alum + CaCO₃ = alum + calcium carbonate; AC+ CaCO₃ = aluminum chloride+ calcium carbonate; PP = Potassium permanganate; SAC = SoftAcid (a mixture of sodium lignosulfonate, formic acid, and propionic acid); SB = sodium bisulfate; COMP = composting; QUI = quicklime; DL = dolomitic limestone; ZEO = zeolite; CH = charcoal; TAR = tarping; ACC = acidified clay; BEN = bentonite; AFC = product containing phosphoric+hydrochloric+citric acids; UN = undefined; 1 = wood shavings; 2 = rice hull; 3 = sawdust; 4 = rice bran; 5 = Chopped elephant; 6 = straw; WG = weight gain; FI = feed intake; FC = feed conversion; Mort = mortality; NH₃ = ammonia; Mo = moisture; Micro = microbiota.

## Mortality

In the mortality rate, there was no statistical difference between the treated group and the untreated group (MD = -0.278, CI = -0.750–0.195, P = 0.249). However, it is possible to see a positive trend in the use of conditioners (Fig 5).

This effect was heterogeneous as indicated by $I^2$ = 84.72%, P<0.001. Subgroup analysis shows that in the group treated with acidifiers there was a significant reduction in the mortality rate when compared to the control group (MD = -0.664, CI = -0.932 - -0.397, P<0.001). In the treatment of alkalizing agents, there was a significant increase in the mortality of broilers (MD = 1,020, IC = 0.648–1,392, P<0.001).

There was no significant effect of the treatment of the litter with gypsum on the mortality rate compared to the control group (P> 0.05), although the graph shows a trend towards a reduction in this variable.

Through this systematic review it was possible to verify that of the 26 studies included in the meta-analysis, only 5 articles evaluated the mortality rate.

## Ammonia

In Fig 6 it is shown that the litter treatment with conditioners reduces the concentration and volatility of ammonia compared to the control group (SMD = -1.014, IC = -1.722 - -0.306, P = 0.005).

This effect was heterogeneous ($I^2$ = 86.39%, P<0.001). In the subgroup analysis there was a significant reduction of ammonia with gypsum utilization (SMD = -6.375, IC = -11.354 - -1.396, P = 0.012) and acidifying (SMD = -1.075, IC = -1.898 - -0.251, P = 0.011) as treatments.

Alkalizing, adsorbents and superphosphate group did not differ statistically from the control group (P>0.05).

## pH

As can be seen in Fig 7, the use of conditioners significantly affects the pH of the litter (MD = -0.293, IC = -0.414 - -0.172, P<0.001). However, this effect was heterogeneous $I^2$ = 93.62%, P<0.001. In the analysis of the subgroups, the acidifiers (MD = -0.579, IC = -0.752 - -0.406, P<0.001) and superphosphate (MD = -0.832, IC = -1.274 - -0.389, P<0.001) showed lower values of pH compared to untreated litter. Alkalizing agents, in turn, had higher pH values (MD = 0.576, IC = 0.042–1,111, P = 0.035). The adsorbents and gypsum (P> 0.05) showed only a tendency to reduce the pH of the litter.

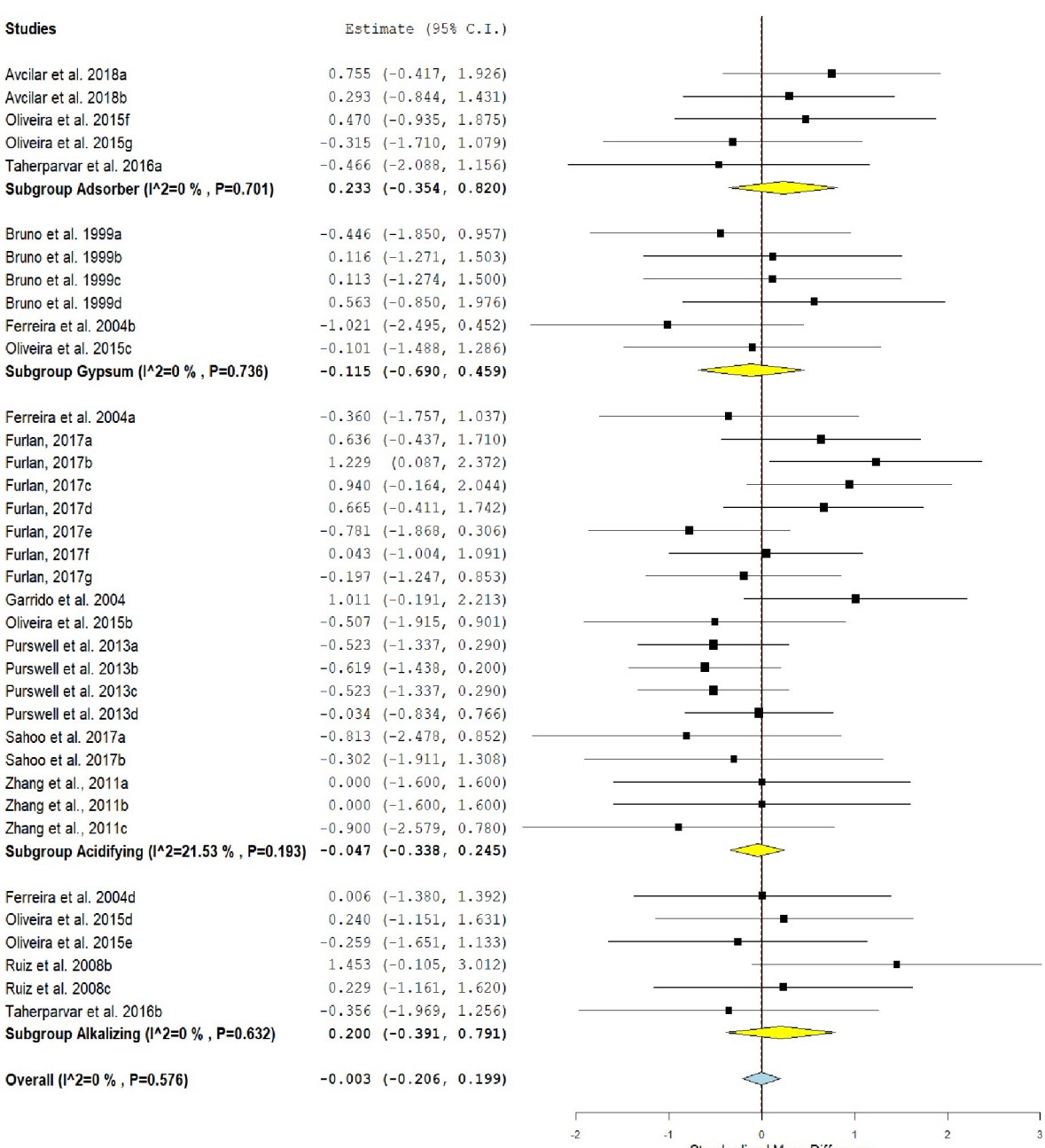

**Fig 2. Forest plot of the effect size or standardized mean difference and 95% confidence interval of the effect of litter treatments on feed intake of broilers.** The solid vertical grey line represents a mean difference of zero, or no effect. Points to the left of the solid vertical line represent a reduction in feed intake, while points to the right of the line indicate an increase in feed intake in broilers reared on treated litter.

## Moisture

Fig 8 shows the overall positive effect of treatments in reducing moisture in litter compared to untreated litter (MD = -2.103, IC = -2.638 - -1.567, P <0.001). This effect was heterogeneous ($I^2$ = 97.72%, P <0.001). The subgroup analysis shows that the use of acidifiers (MD = -6,438, IC = -10,416 - -2,460, P = 0.002), adsorbents (MD = -4,959, IC = -8,562 - -1,375, P = 0.007) and alkalizing agents (MD = -1.253, IC = -1.849 - - 0.658, P <0.001) reduced the moisture

content of the litter. Gypsum showed only a leak tendency to reduce litter moisture compared to untreated litter (MD = -3.503, IC = -9.821–2.816, P = 0.277).

## Pathogenic microbiota

The use of conditioners in the poultry litter demonstrated a positive overall effect (Fig 9) for the reduction of the pathogenic microbiota (SMD = -1,457, IC = -2,118 - -0,796, P<0.001).

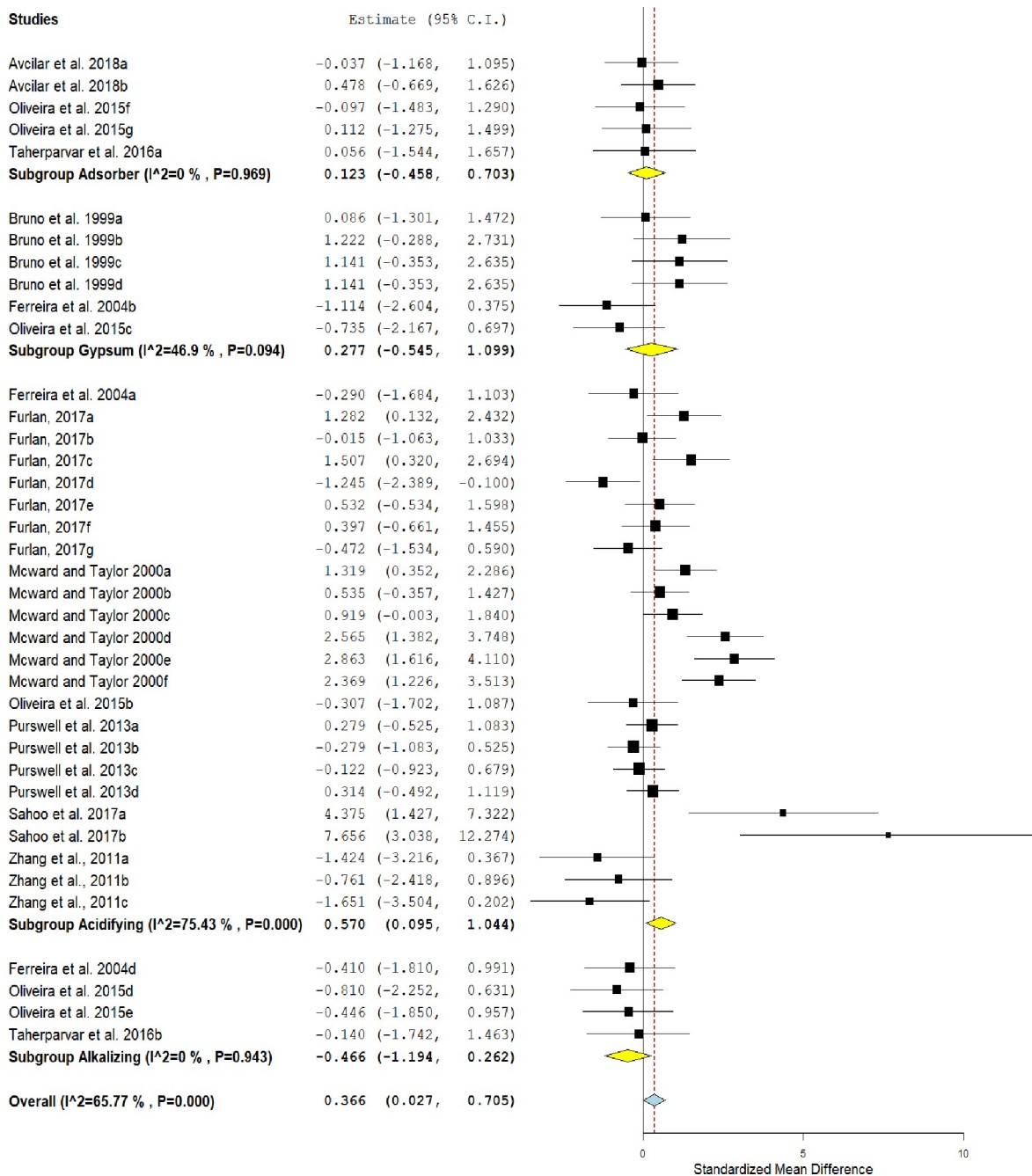

**Fig 3. Forest plot of the effect size or standardized mean difference and 95% confidence interval of the effect of litter treatment on weight gain of broilers.** The solid vertical grey line represents a mean difference of zero, or no effect. Points to the left of the solid vertical line represent a reduction in weight gain, while points to the right of the line indicate an increase in the weight gain of broilers reared on treated litter.

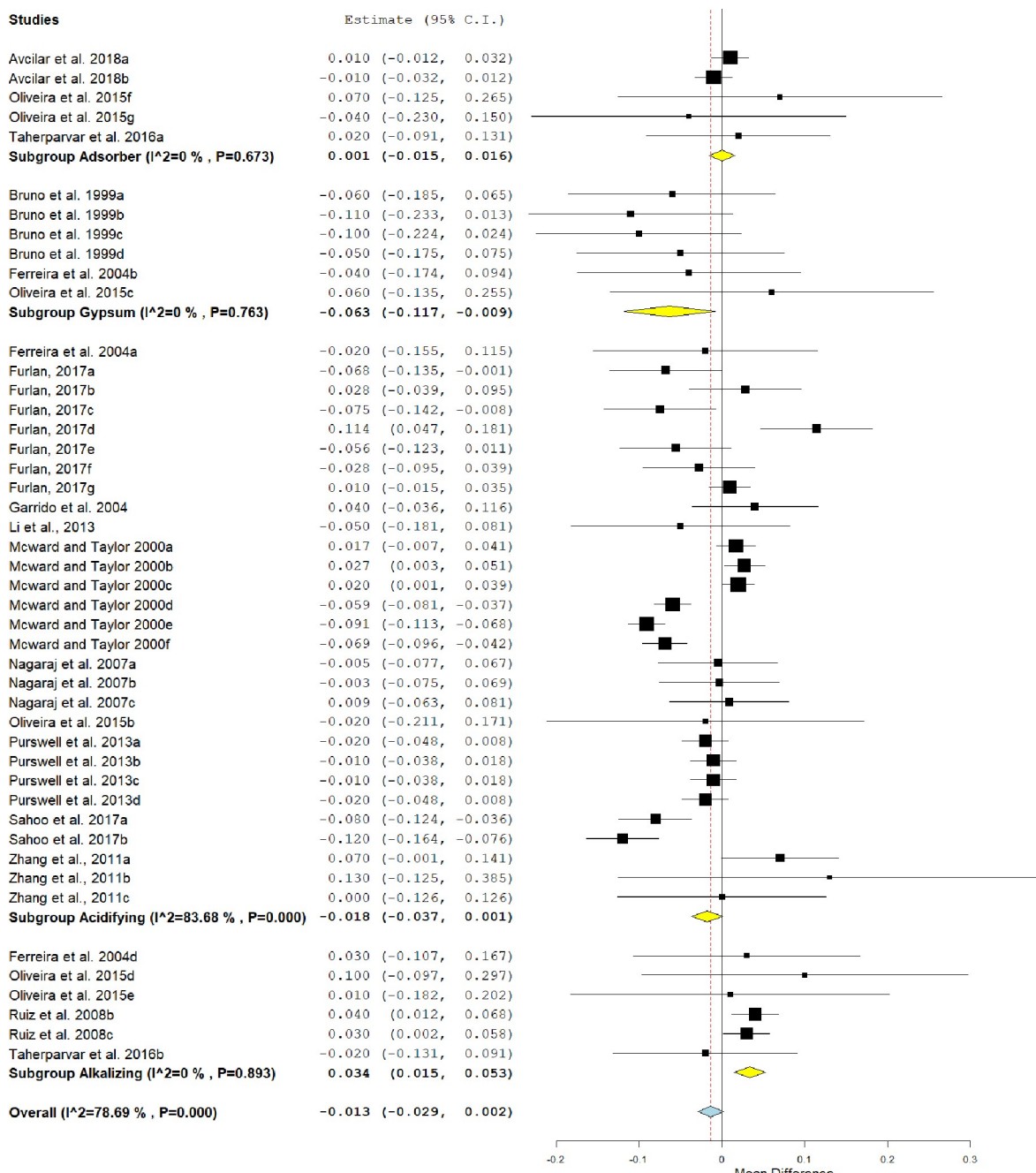

**Fig 4. Forest plot of the effect size or mean difference and 95% confidence interval of the effect of litter treatment on feed conversion of broilers.** The solid vertical grey line represents a mean difference of zero, or no effect. Points to the left of the solid vertical line represent an improvement in feed conversion, while points to the right of the line indicates a worsening in feed conversion of broilers reared on treated litter.

This effect was heterogeneous ($I^2$ = 68.44%, P<0.001). In the subgroup analysis, acidifiers (SMD = -1,214, IC = -1,866 - -0,562, P<0.001) and alkalizers (SMD = -1,753 IC = -2,792 - -0,713 P<0.001) also promoted a reduction in the microbiota pathogenic. Gypsum and adsorbents showed a positive trend in reducing the pathogenic microbiota in treated litters.

## Publication bias

The symmetry of the funnel plots (Fig 10A, 10B, 10C, 10E, 10F and 10G), demonstrated statistically by the non-significance of Egger's linear regression test, showed that the results of feed intake, weight gain, feed conversion, mortality rate, pH and moisture, respectively, were not affected by risk of publication bias. Unlike, as shown in Fig 10D the funnel plot is asymmetric with studies with larger standards errors and positive effect size seems to be missing, therefore evidencing publication bias for mortality rate. For this response variable some studies could not be included in the meta-analysis as they presented the survival rate instead of the mortality rate. These two response variables have opposite interpretations and for this reason they cannot be grouped in the same meta-analysis. Egger test revealed some outliers, represented by dots outside of the funnel plots, as shown in Fig 10. The outliers are due to high heterogeneity between studies. As shown in Fig 10H it seems that studies with a positive effect size for pathogenic microbiota response are missing.

## Discussion

### Literature search and extracted information

This systematic literature search shows that 31% of the studies included in this meta-analysis are very recent, having been published from 2019 to 2015 and 42% of them are recent (2014–2005) and 27% not so recent (2004–1999).

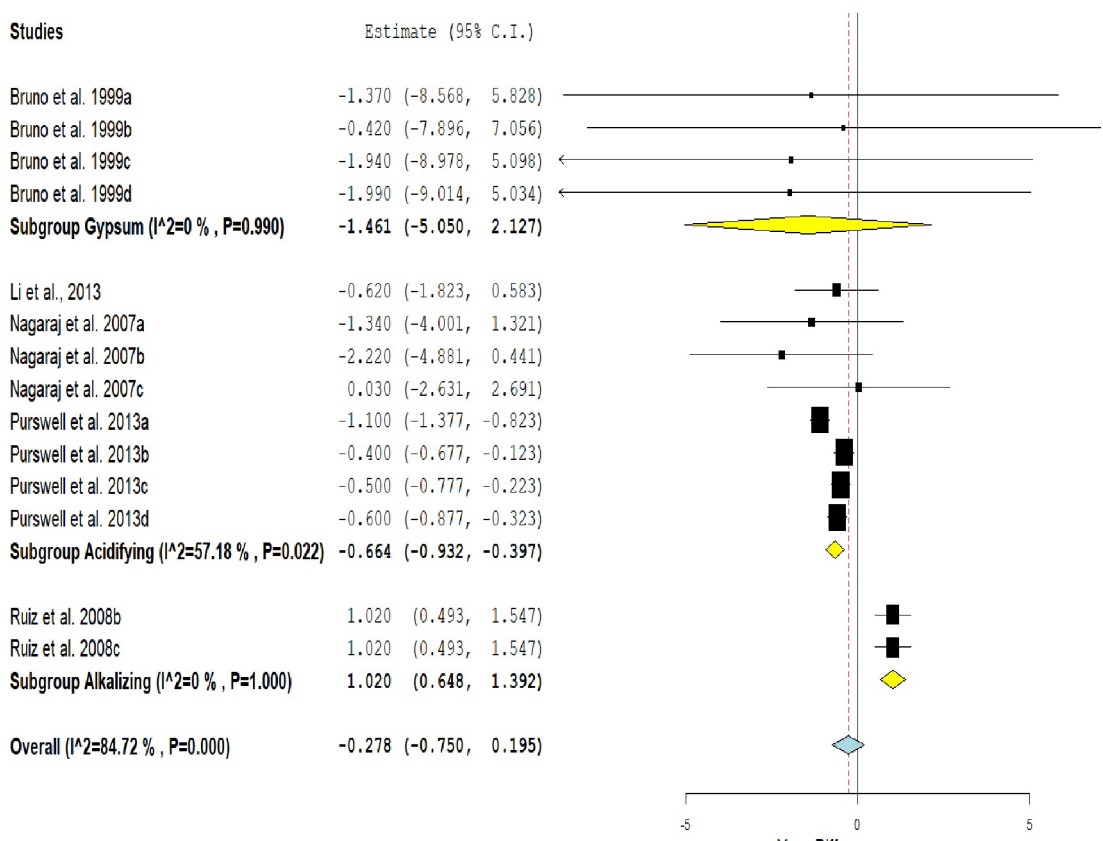

**Fig 5. Forest plot of the effect size or mean difference and 95% confidence interval of litter treatment on mortality rate in broilers.** The solid vertical grey line represents a mean difference of zero, or no effect. Points to the left of the line represent a reduction in mortality, while points to the right of the line indicate an increase in mortality in broilers reared on treated litter.

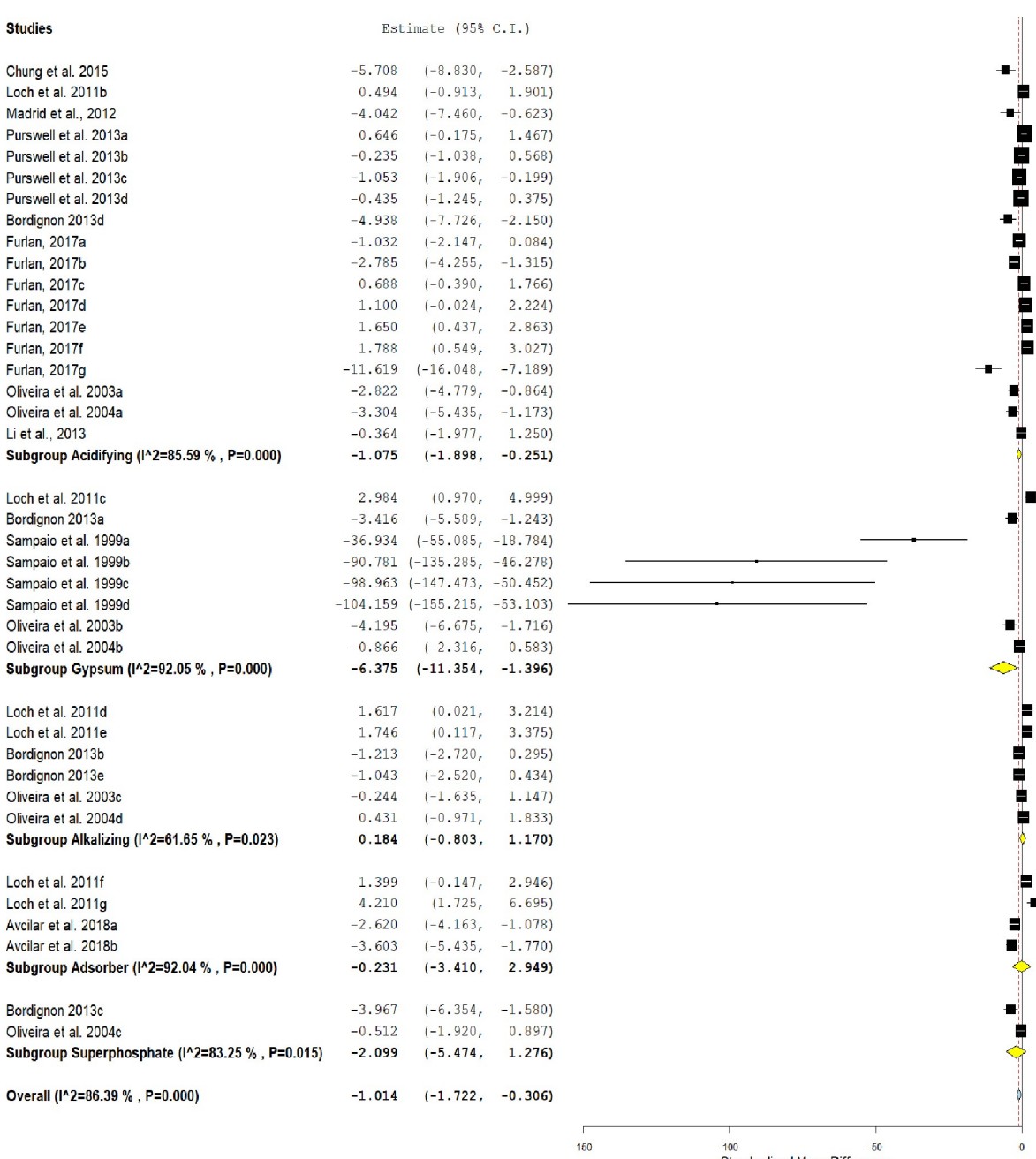

**Fig 6. Forest plot of the effect size or standardized mean difference and 95% confidence interval of litter treatment on ammonia concentration and volatization.** The solid vertical grey line represents a mean difference of zero, or no effect. Points to the left of the line represent a reduction in ammonia, while points to the right of the line indicate an increase in ammonia on treated litter.

This study presents results obtained from 10 countries representing the European, Asian and American continents, each one, with its characteristics of culture, technology and management, making it difficult to describe the reality of each region.

Brazil presents the largest number of studies (seven) followed by the United States of America (USA) with six studies. These results, interestingly, coincide with the fact that Brazil and the United States are respectively, the first and second largest chicken meat exporters in the

world, reflecting the importance of this research topic for these two countries. Despite this, the studies referenced include emerging countries from a broiler production point of view, such as India, Spain, Turkey and South Korea.

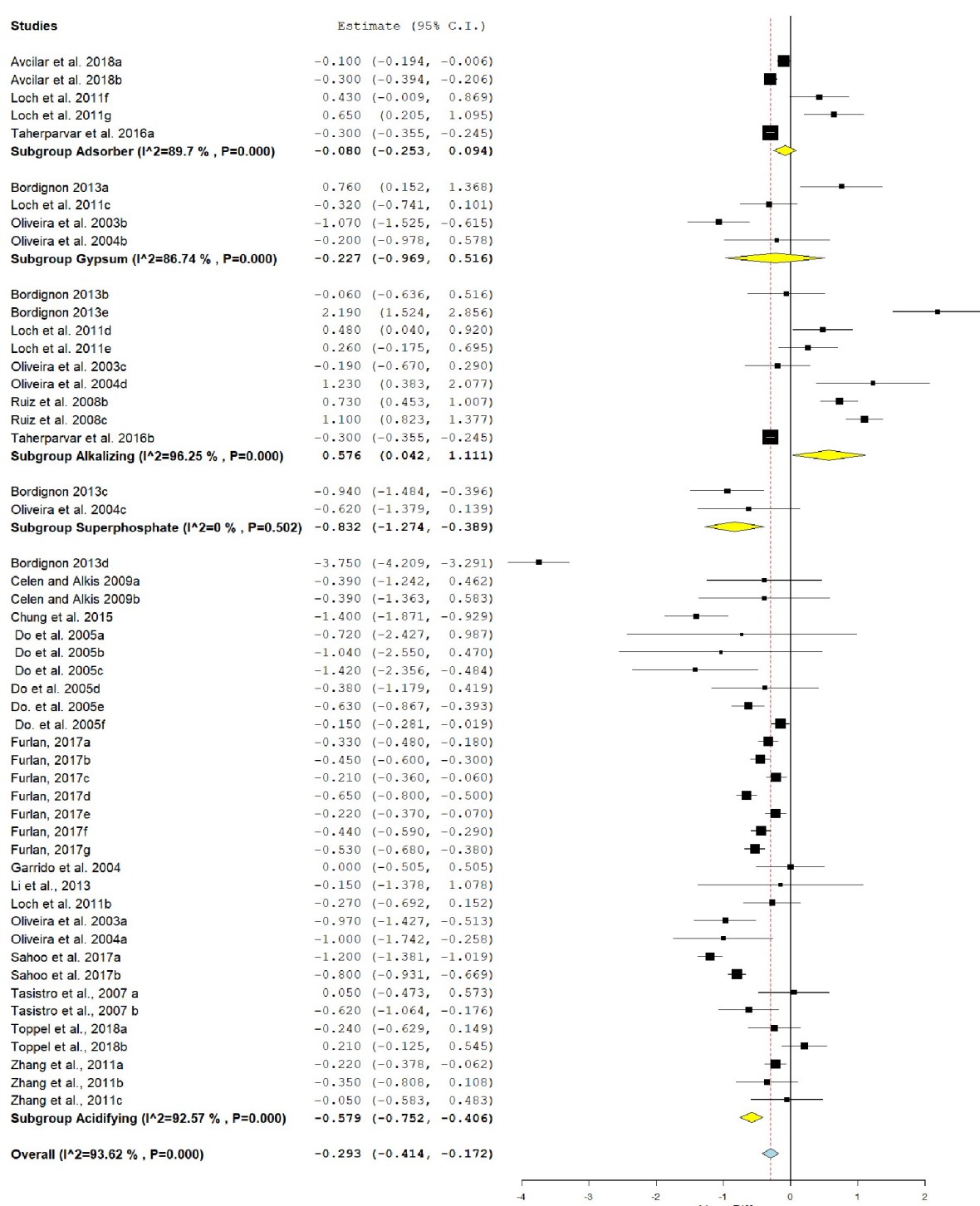

**Fig 7. Forest plot of the effect size or mean difference and 95% confidence interval of litter treatment on pH.** The solid vertical grey line represents a mean difference of zero, or no effect. Points to the left of the line represent a reduction in pH, while points to the right of the line indicate an increase in pH on treated litter.

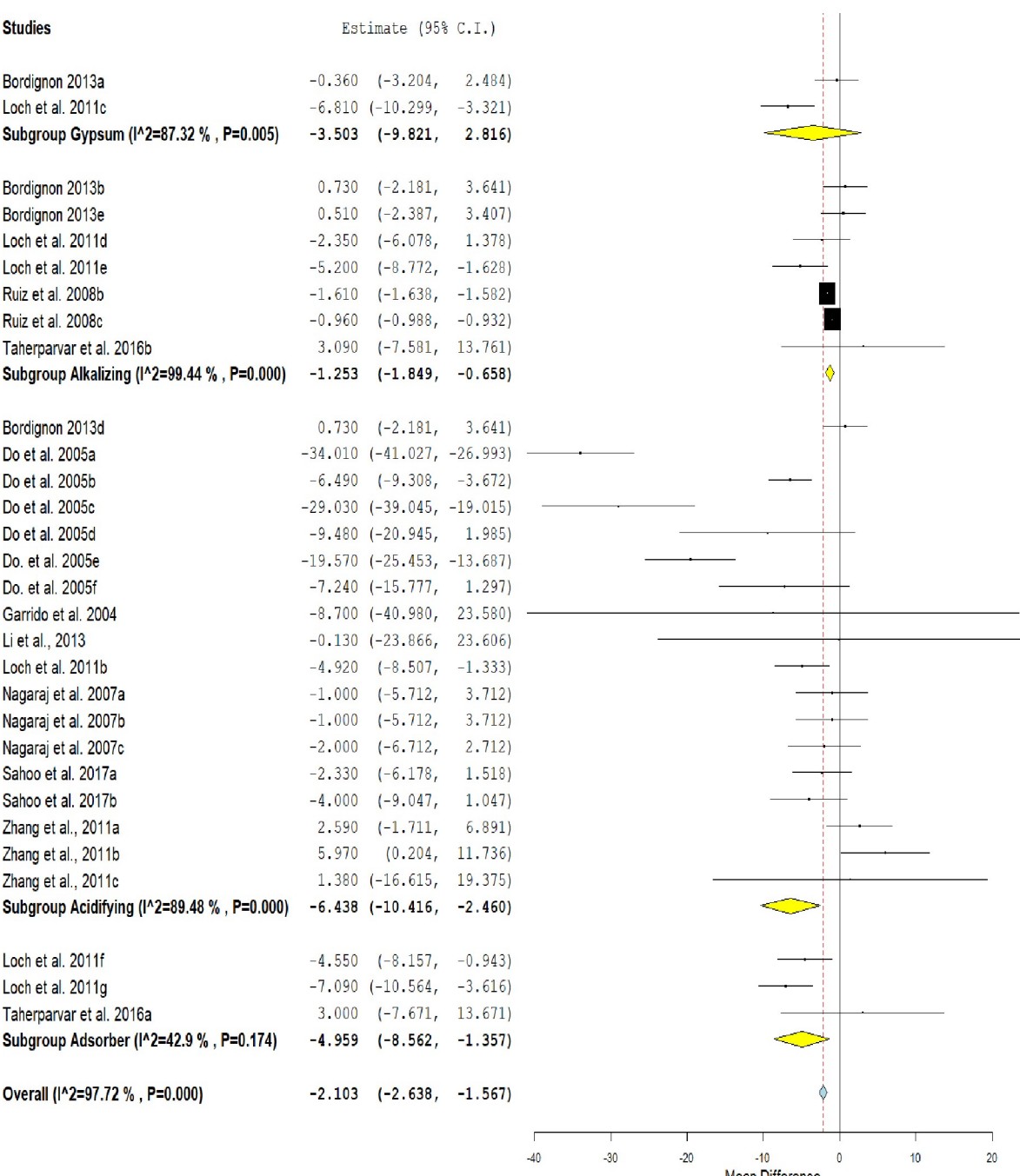

**Fig 8. Forest plot of the effect size or mean difference and 95% confidence interval of litter treatment on moisture.** The solid vertical grey line represents a mean difference of zero, or no effect. Points to the left of the line represent a reduction in moisture, while points to the right of the line indicate an increase in moisture on treated litter.

Of the total of studies included in the meta-analysis 23 different treatments were evaluated (Table 1). It is possible to observe that the most studied litter treatment in the world uses acidifying substances that aim to improve its quality by significantly reducing its pH.

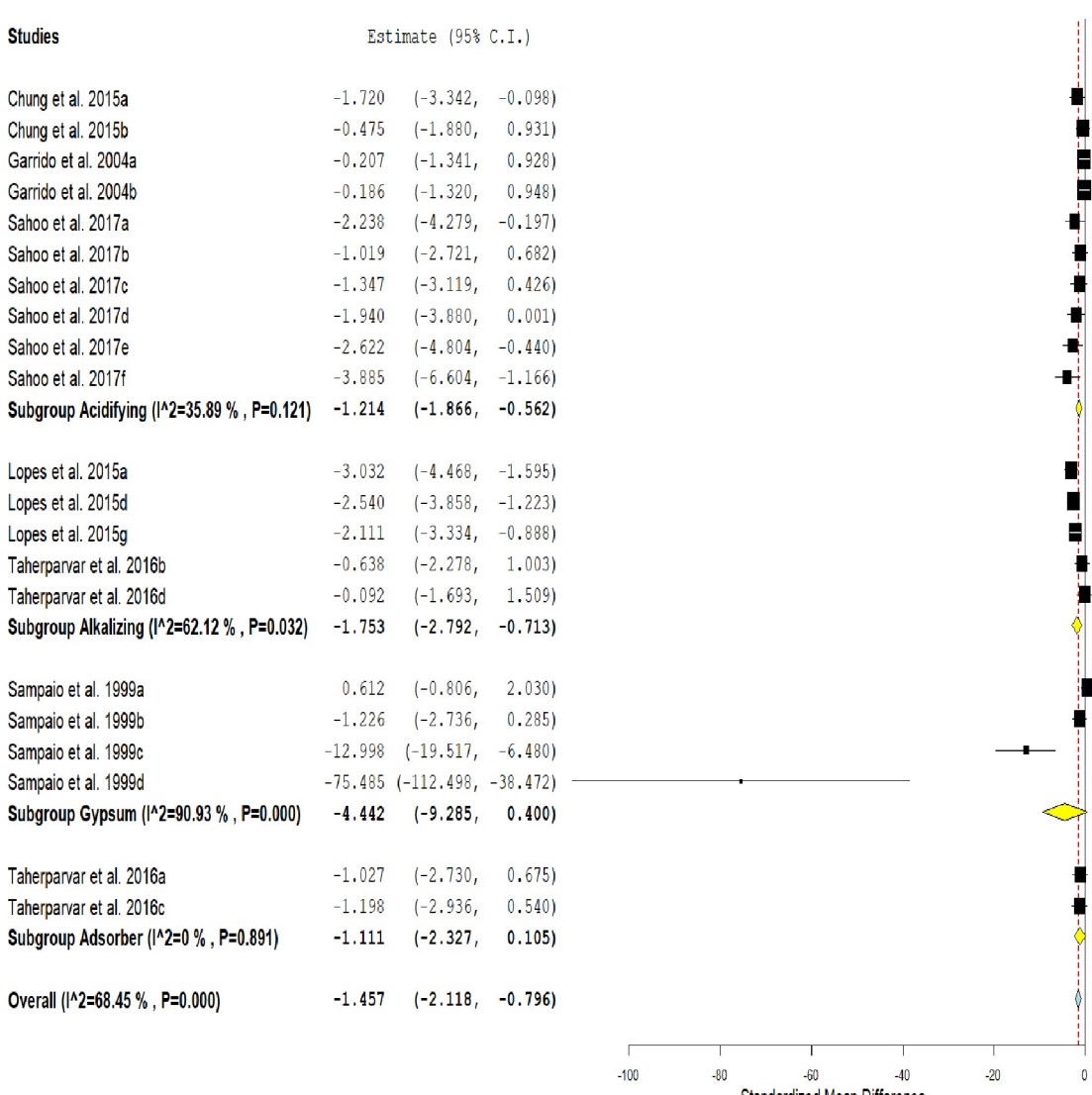

**Fig 9. Forest plot of the effect size or standardized mean difference and 95% confidence interval of litter treatment on pathogenic microbiota.** The solid vertical grey line represents a mean difference of zero, or no effect. Points to the left of the line represent a reduction in pathogenic microbiota, while points to the right of the line indicate an increase in pathogenic microbiota on treated litter.

## Summary of main results on litter quality and broiler performance

The overall findings of the meta-analysis demonstrate that using litter conditioners is beneficial for litter quality characteristics (Figs 6, 7, 8 and 9) weight gain (Fig 3). However, this finding is associated with a moderate degree of heterogeneity.

Therefore, the hypothesis of superiority of the litter treatment must be analyzed with caution, because depending on the evaluated characteristic, the treatments can have contradictory results, but they can be equally satisfactory for the chicken producers, as what happens, for example, in the effect of acidifying and alkalizing on litter pH.

Treatment with conditioners can reduce ammonia volatility and increase nitrogen fixation in the litter through chemical reactions [45, 55]. Ammonia emission increases when factors such as temperature and humidity are high [56]. On the other hand, it is observed that

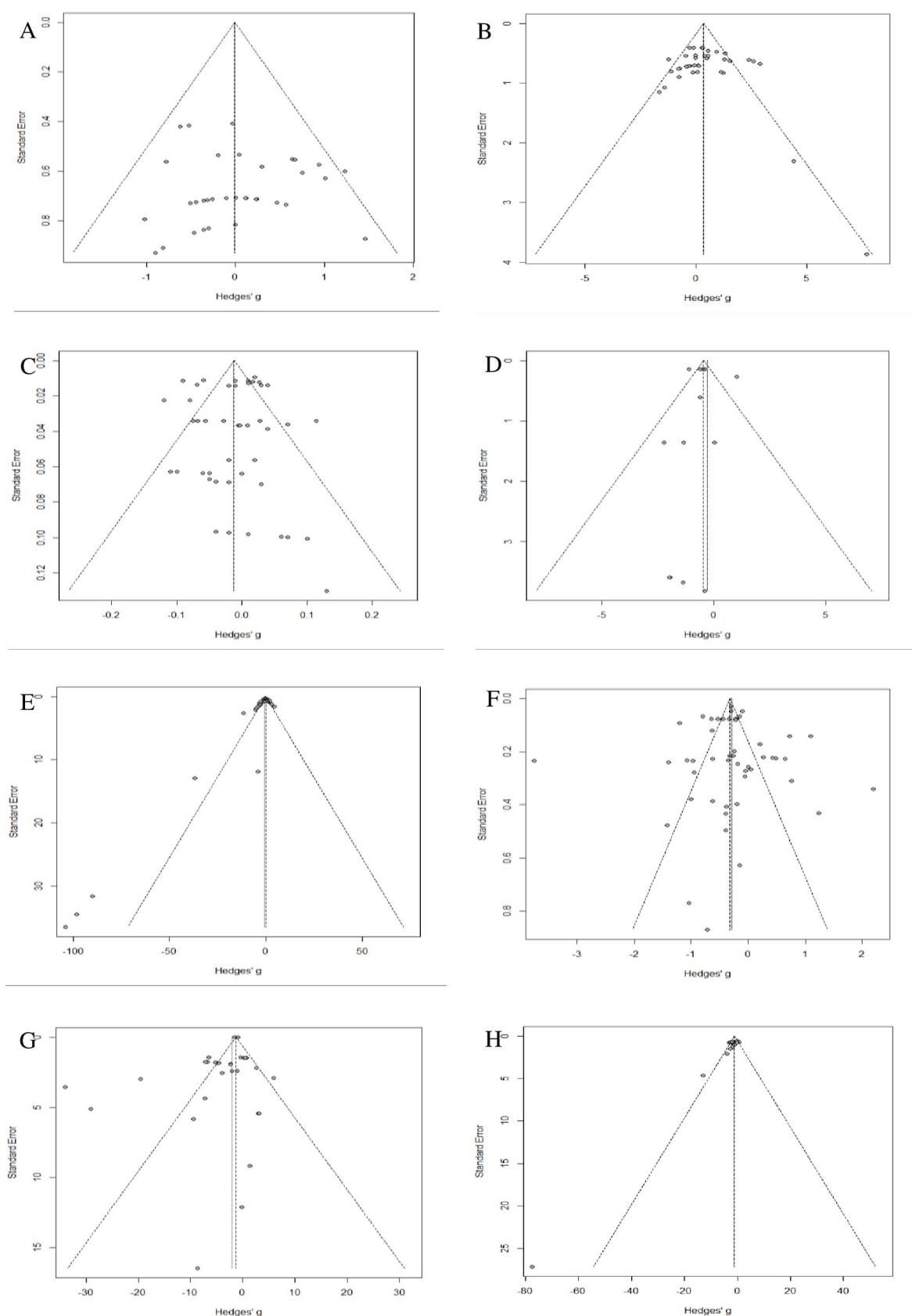

**Fig 10. Publication bias analysis, funnel plot.** Egger's Linear regression test of funnel plot asymmetry: A) Feed intake: bias±s.e. = 0.43 ± 0.68; slope = -0.27 (t = 0.63, p-value = 0.53). B) Weight gain: bias±s.e. = 0.29 ± 0.82; slope = 0.16 (t = 0.36, p-value = 0.72). C) Feed conversion: bias±s.e. = -0.06 ± 0.54; slope = -0.01 (t = -0.11, p-value = 0.91). D) Mortality rate: bias±s.e. = 0.36 ± 1.0; slope = -0.52 (t = 0.36, p-value = 0.72). E) Amonnia: bias±s.e. = -2.26 ± 0.6; slope = 1.27 (t = -3.76, p-value < 0.001). F) pH: bias±s.e. = 0.09 ± 0.81; slope = -0.32 (t = 0.12, p-value = 0.91). G) Moisture: bias±s.e. = -1.04 ± 1.22; slope = -1.27 (t = -0.86, p-value = 0.39). H) Pathogenic microbiota: bias±s. e. = -2.50 ± 0.65; slope = 0.92 (t = -3.85, p-value = 0.001).

ammonia emissions correlate positively with the pH of the litter [57]. Similarly, ammonia volatilization is positively related to moisture [58]. Litter pH plays an important role in ammonia volatilization. Once formed, the free ammonia will be in one of two forms: $NH_3$ without charge or in the form of ammonium ion ($NH_4^+$), depending on the pH of the litter, as the ammonia concentration increases with increasing pH. The release of ammonia is lower when the pH of the litter is below 7.0, but is greater when it is above 8.0, with the decomposition of uric acid being favored under conditions of alkaline pH [55].

Thus, the litter has a special condition for bacterial development with pH values between 8 and 9 in reused litters, and water activity, between 0.90 and 0.92 [59]. The litter offers conditions for the development of many undesirable bacteria, such as, for example, *Salmonella* spp., *Campylobacter* spp., *E. coli*, *Clostridium perfringens* and *Staphylococcus aureus* [60]. The microbiota of the litter is extremely diverse due to the continuous supply of fecal material during the rearing cycle, in addition to the incorporation of fungi and bacteria derived from the environment [61]. In this context, the concept that the simple fecal accumulation in the litter results in an increase in pathogenic microorganisms is very common, in addition to intensifying the generation of harmful gases to the health of birds [62]. In addition, the reuse of the litter increases the moisture content and the denitrifying bacteria that intensify the production of ammonia [63]. Ammonia is a colorless and irritating gas for mucous membranes that affect both birds and people involved in the management of chickens. Ammonia when inhaled in amounts greater than 25 ppm causes weight loss in birds [64] and above 60 ppm predisposes birds to respiratory problems, which ends up promoting the increase of secondary complications after vaccinations, and the rate and depth of breathing may also be reduced, thus impairing the physiological processes of gas exchange, in addition to contributing to the carcass declassification due to skin lesions [45]. The development of ocular problems in birds also stands out [65]. High levels of ammonia depress performance and worsen broiler feed conversion [66]. Ammonia negatively affects broiler growth and performance [67]. Ammonia reduction promotes better performance of birds under commercial conditions [68].

Thus, we can consider that there is a chain reaction of the treatments in each analyzed variable, since, in the meta-analysis of the subgroups, the treatments act directly on the pH, moisture and pathogenic microbiota reflecting on the concentration and volatility of ammonia, which will reflect on performance parameters, affecting weight gain, feed conversion and mortality rate.

It is important to note, however, that feed intake was the only variable not to be influenced by treatments (Fig 2). It may be inferred that probably the types of litter conditioners did not have a sufficient effect to change eating behavior. This result can possibly be explained by the basic behavioral needs of birds that prioritize the consumption of food for survival that is not altered within normal limits of litter quality.

Among the subgroups, acidifiers were shown to have a positive effect for all variables analyzed. The acidifying agents act by decreasing the pH and inhibiting the bacterial action in the conversion of nitrogen to ammonia [66]. This reduction in pH improves conditions inside the facilities, since ammonia only volatilizes under alkalinity conditions [69]. In addition, acidifiers contribute to the inactivation of pathogenic bacteria by creating an unfavorable environment for their development [37, 63]. These characteristics, associated with a greater number of

publications found, are probably responsible for the positive results obtained by this treatment in the meta-analysis. Within the group of acidifiers, aluminum sulfate was the most studied substance. Aluminum sulfate produces 6 moles of $H^+$ when it dissolves, the $H^+$ produced by this reaction reacts with $NH_3$ to form $NH_4^+$, which can react with the sulfate ions, forming 3 moles of ammonium sulfate which is soluble in water [70].

Several authors report the positive effect of aluminum sulfate in reducing pH, moisture, ammonia volatility and pathogenic microbiota [14, 37, 43, 46, 71, 72, 73, 74, 75, 76]. Other authors have also reported the positive effect of aluminum sulfate on the performance characteristics of birds [43, 45, 70, 77]. In addition, the literature also describes other acidifiers as valid and effective options for improving the characteristics of litter and broiler performance, such as: propionic acid, monobasic calcium phosphate, phosphoric acid, iron sulfate [44], chloride aluminum [78], citric acid [79], sodium bisulfate [41, 47, 80] and alum [28, 38, 71,74, 78, 81, 82].

However, it is important to note that acidifiers have difficulty in application, especially in aviaries of family labor, because they need more sophisticated protective equipment and because they present risks to the applicator due to the low pH of the product.

Gypsum ($CaSO_4$) used to treat poultry litter increases nitrogen fixation through chemical reactions, thus avoiding an increase in the concentration of ammonia in the environment [35, 45, 55].

In the meta-analysis, this effect was evidenced, as well as, an improvement in feed conversion was demonstrated. Although, no significant difference was found in the comparison with untreated litter, gypsum showed a positive tendency to reduce pH, pathogenic microbiota and consequently mortality, in addition to increasing weight gain. A possible explanation for the absence of significant effects of gypsum in the present meta-analysis could be due to the small number of studies published to date, which ends up reducing the power of the test. For this reason, the need to expand studies on the use of gypsum in poultry litter becomes evident, as the results suggest positive results with its application.

Litters treated with different doses of gypsum (10, 20, 30 and 40%) showed lower pH values [50] and reduced ammonia [45, 83, 84, 85]. On the other hand, a study found that the increase in moisture in litter treated with gypsum, leads to a decrease in ammonia volatilization, without major changes in pH values [86].

These authors attribute the result to the high dissociative affinity of ammonia in water. This was not observed in the present meta-analysis because, gypsum showed a tendency to reduce the moisture content. Another study reported a significant decrease in the standard microorganism count due to the decrease in ammonia volatilization, due to the accumulation of fecal mass and, mainly, to the lower amount of ammonia released by the litter due to the incorporation of gypsum [50].

The alkalizing subgroup, in turn, has a characteristic of raising the pH during the fallowing time. Although alkaline substances have a negative effect on high levels inside the house for causing discomfort to birds, it is extremely important for the control of some populations of microorganisms that develop in the litter.

This alkalinity condition creates an unfavorable environment for bacterial development, in addition to promoting a rapid volatilization of ammonia in a period when the aviary is closed and without birds inside [1]. Furthermore, alkalizers are also used as a strategy to control moisture. In the meta-analysis, as expected, alkalizers promoted an increase in pH, a reduction in moisture content and a significant reduction in the pathogenic microbiota of the litter. Alkalizing agents raise the pH, making the environment inhospitable for bacteria, in addition to acting by reducing the amount of free water and decreasing the water activity of the poultry litter [13, 17, 59]. Water activity directly influences the survival of microorganisms. With less

water available, the microorganism will need more energy to remove it from the litter, in order to use it in its metabolism, hindering or preventing its survival [87].

However, alkalizers seem to have a negative influence on the performance of birds, since they showed significantly worse feed conversion compared to untreated litter and, also, a negative trend for weight gain and mortality rate. Several authors have not found a significant effect of the application of lime on the performance of broilers [15, 48, 51, 88,89].

Although the meta-analysis has shown some negative points in the performance of broilers, the use of alkalizers has a number of advantages according to the practical experience of the authors of this review. One of the advantages attributed to the use of lime is the final cost applied (transport, logistics and application), which is around USD 78.0 a ton. Another advantage of the lime is the ease of application that does not require specialized labor or sophisticated equipment. The producer himself is responsible for the treatment of the litter. The risks associated with its use are also lower because it presents a low physic-chemical risk of intoxication for the applicator as long as following the rules for using basic protective equipment (common mask, gloves, boots and overalls). In addition, lime improves the physicochemical properties of the litter as a future fertilizer, especially for acidic soils incorporating calcium into the soil. It is evident then, the need for further studies to elucidate the real effect of alkalizing agents on the performance of birds.

The adsorbents work by reducing the release of ammonia and adsorbing moisture [66]. Litter conditioners with a high capacity to adsorb moisture, reduce the activity of ammonia-producing bacteria, and therefore the pH of the litter [45]. The adsorption of water occurs through the hydration of cations that compensate the surface load by osmotic balance [90].

The results found in the meta-analysis demonstrated this effectiveness in reducing moisture. For the other response variables, it was possible to notice a positive trend to increase weight gain and reduce pH, ammonia volatility and pathogenic microbial load.

These results are corroborated by studies [15, 34, 51,91], which despite not having found significant difference in performance, showed that the adsorbents improved the quality of the litter. By normalizing the litter moisture, the adsorbents control the pathogenic microbiota [51]. Therefore, the results indicate that adsorbents can influence litter characteristics without affecting broiler performance.

Simple superphosphate has an inhibitory action on litter ammonia volatilization, being one of the oldest recommendations to inhibit ammonia losses from organic waste [85]. There is an association of the mode of action of superphosphate with aluminum sulfate because it has acidic characteristics acting on the reduction of pH, microbiological activity and ammonia volatilization [92]. Although superphosphate in this meta-analysis was able to reduce the pH, it did not influence ammonia. This may have occurred due to the small amount of existing publications (two articles) in this subgroup since the forest plot shows a positive trend in reducing ammonia. For this reason, the results obtained in this comparison should be interpreted with caution.

## Unconventional treatments

Despite not being included in the meta-analysis due to the lack of studies, non-conventional treatments also deserve to be discussed as possible new lines of research. In this sense, the interpretation of the results found in the literature should be done with caution as there is a small number of studies. One of the litter treatments still not widespread is the use of enzymatic biocatalyst. Results indicate that the performance was not affected with Rydall (enzyme biocatalyst) and Myzuho (bacterial culture) [30]. The authors attribute this result to the fact that the magnitude of the reduction in the level of ammonia in the Rydall group was only

12.5% compared to the control and, therefore, may not have been strong enough to evoke a positive response in birds.

Another unconventional treatment is the use of non-absorbent polymers. In one study, no significant differences were found in the performance of broilers reared on litter treated with non-absorbent polymers [32].

Another unconventional treatment in the production of broilers is the use of *Y. shidigera* extract. Authors reported that there was no statistical difference in the values of pH, moisture and ammonia between litters treated with *Y. shidigera* extract and untreated litters [31].

The authors argue that the effect of the treatment in question could be better observed when used in litter in poor condition. However, in order to be considered a safe alternative, further studies are needed to prove its effectiveness.

The publication bias analyses showed that publication bias had little impact in most response variables, thus increasing confidence in the results of this review. In cases where publication bias analyses indicate that bias may exist (mortality rate and pathogenic microbiota) it is important to take into account that comparing a litter treatment with nontreatment, can lead to an overestimation of the effect size of treatment under investigation. Despite of this bias risk, the meta-analysis showed that the litter must be treated properly to maintaining or even improving the performance of chickens.

## Conclusions

Acidifiers are associated with a reduction in pH, ammonia, moisture and pathogenic litter microbiota, with consequent greater weight gain and lower mortality rate for broilers. Gypsum is effective in reducing ammonia concentration and volatility and improving feed conversion.

Alkalizing agents increase pH, reduce moisture and pathogenic microbiota, but worsen feed conversion and increase the mortality rate. Superphosphate lowers pH while adsorbents reduce litter moisture. None of the litter treatments influence the feeding behavior of birds.

## Supporting information

**S1 Table. Data for Feed intake meta-analysis.**
(DOCX)

**S2 Table. Data for weight gain meta-analysis.**
(DOCX)

**S3 Table. Data for feed conversion meta-analysis.**
(DOCX)

**S4 Table. Data for mortality rate meta-analysis.**
(DOCX)

**S5 Table. Data for ammonia concentration meta-analysis.**
(DOCX)

**S6 Table. Data for pH meta-analysis.**
(DOCX)

**S7 Table. Data for moisture meta-analysis.**
(DOCX)

**S8 Table. Data for pathogenic microbiota meta-analysis.**
(DOCX)

## Author Contributions

**Conceptualization:** Taiani dos Santos de Toledo, Marcos Antonio Dai Prá, Fábio Pereira Leivas Leite, Victor Fernando Büttow Roll.

**Data curation:** Aline Arassiana Piccini Roll.

**Formal analysis:** Taiani dos Santos de Toledo, Aline Arassiana Piccini Roll, Victor Fernando Büttow Roll.

**Funding acquisition:** Victor Fernando Büttow Roll.

**Methodology:** Aline Arassiana Piccini Roll, Victor Fernando Büttow Roll.

**Project administration:** Victor Fernando Büttow Roll.

**Supervision:** Aline Arassiana Piccini Roll, Fernando Rutz, Henrique Müller Dallmann, Fábio Pereira Leivas Leite, Victor Fernando Büttow Roll.

**Writing – original draft:** Taiani dos Santos de Toledo.

**Writing – review & editing:** Aline Arassiana Piccini Roll, Fernando Rutz, Henrique Müller Dallmann, Marcos Antonio Dai Prá, Fábio Pereira Leivas Leite, Victor Fernando Büttow Roll.

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
