## [Decision Letter · Decision Letter 0]

5 Mar 2020

PONE-D-20-02805

An assessment of the impacts of litter treatments on the litter quality and broiler performance: a systematic review and meta-analysis

PLOS ONE

Dear Prof. Roll,

Thank you for submitting your manuscript to PLOS ONE. After careful consideration, we feel that it has merit but does not fully meet PLOS ONE’s publication criteria as it currently stands. Therefore, we invite you to submit a revised version of the manuscript that addresses the points raised during the review process.

The manuscript should be revised deeply. The manuscript should be presented according to guidelines for authors of Plos One. Thank you for your hard work.

We would appreciate receiving your revised manuscript by Apr 19 2020 11:59PM. To enhance the reproducibility of your results, we recommend that if applicable you deposit your laboratory protocols in protocols.io, where a protocol can be assigned its own identifier (DOI) such that it can be cited independently in the future. For instructions see: http://journals.plos.org/plosone/s/submission-guidelines#loc-laboratory-protocols

We look forward to receiving your revised manuscript.

Kind regards,

Arda Yildirim, Ph.D.

Academic Editor

PLOS ONE

Additional Editor Comments (if provided):

The study is very well presented, I feel that the manuscript is dealing with a good topic but lacks in the quality of preparation. Please review the referee comments and make your peer revision.

Journal Requirements:

2)  Thank you for stating the following in the Competing Interests section:

[The authors have declared that no competing interests exist.].   

We note that one or more of the authors are employed by a commercial company: Brasil Foods S/A.

i. Please provide an amended Funding Statement declaring this commercial affiliation, as well as a statement regarding the Role of Funders in your study. If the funding organization did not play a role in the study design, data collection and analysis, decision to publish, or preparation of the manuscript and only provided financial support in the form of authors' salaries and/or research materials, please review your statements relating to the author contributions, and ensure you have specifically and accurately indicated the role(s) that these authors had in your study. You can update author roles in the Author Contributions section of the online submission form.

ii. Please also provide an updated Competing Interests Statement declaring this commercial affiliation along with any other relevant declarations relating to employment, consultancy, patents, products in development, or marketed products, etc. 

Reviewers' comments:

Reviewer's Responses to Questions

**Comments to the Author**

1. Is the manuscript technically sound, and do the data support the conclusions?

Reviewer #1: Yes

Reviewer #2: Yes

2. Has the statistical analysis been performed appropriately and rigorously? 

Reviewer #1: Yes

Reviewer #2: Yes

3. Have the authors made all data underlying the findings in their manuscript fully available?

Reviewer #1: Yes

Reviewer #2: Yes

4. Is the manuscript presented in an intelligible fashion and written in standard English?

Reviewer #1: Yes

Reviewer #2: Yes

5. Review Comments to the Author

Reviewer #1: In general, the manuscript clearly presents a straightforward meta-analysis, although the number of studies entering the meta-analysis is somewhat on the low side. There are some very occasional issues with English language idiomatic usage, however, overall the writing is quite acceptable.

The following are generally minor comments:

1) Line 147: It is a bit unclear what the authors mean by "there were no repetitions of publications". Does this mean that there was only a single publication covering this treatment?

2) From the PRISMA checklist: "Describe all information sources (e.g., databases with dates of coverage, contact with study authors to identify additional studies) in the search and date last searched."

Did the authors contact study authors?

3) From the PRISMA checklist: "Describe methods used for assessing risk of bias of individual studies (including specification of whether this was done at the study or outcome level), and how this information is to be used in any data synthesis."

This was not performed.

Reviewer #2: 1- The authors rigorously explain their research criteria and the conclusions are drawn appropriately based on the data presented.

2- The data criteria and data analyses were thoroughly explained and support the conclusions.

3- The authors have provided supplemental materials that that support their findings.

4- The manuscript has minor grammatical errors that need to be fixed, but it is written in an orderly fashion.

6. PLOS authors have the option to publish the peer review history of their article (what does this mean?). If published, this will include your full peer review and any attached files.

Reviewer #1: No

Reviewer #2: No

---

## [Author Response · Author response to Decision Letter 0]

30 Mar 2020

March 30, 2020

Dear Dr. Arda Yildirim and Reviewers

First of all, we wish to express our appreciation for your comments, suggestions, and corrections, which have improved the manuscript. In this way, we inform you that all of them have been taken into account, and are detailed below:

Journal Requirements:

2) Thank you for stating the following in the Competing Interests section:

[The authors have declared that no competing interests exist.]. 

We note that one or more of the authors are employed by a commercial company: Brasil Foods S/A.

i. Please provide an amended Funding Statement declaring this commercial affiliation, as well as a statement regarding the Role of Funders in your study. If the funding organization did not play a role in the study design, data collection and analysis, decision to publish, or preparation of the manuscript and only provided financial support in the form of authors' salaries and/or research materials, please review your statements relating to the author contributions, and ensure you have specifically and accurately indicated the role(s) that these authors had in your study.

ii. Please also include the following statement within your amended Funding Statement. 

iii. “The funder provided support in the form of salaries for authors [insert relevant initials], but did not have any additional role in the study design, data collection and analysis, decision to publish, or preparation of the manuscript. The specific roles of these authors are articulated in the ‘author contributions’ section.”

iv. Please include both an updated Funding Statement and Competing Interests Statement in your cover letter. We will change the online submission form on your behalf.

Thanks editor for the comments provided and for helping us with the online submission forms. We have addressed all the comments and updated the manuscript with our statements relating to the author contributions, funding and competing interests. We hope that now all style standards have been met.

Author contributions

VFBR, FPLL, MADP and TST conceived of the presented idea. VFBR, FPLL and AAPR supervised the project. TST and AAPR carried out the literature search and data extraction, performed the analytic calculations and processed the experimental data. VFBR performed the analysis and designed the figures. FR, HMD and MADP helped supervise the project and provided critical feedback to the writing of the manuscript. All authors discussed the results and contributed to the final manuscript.

Funding statement

The Brasil Foods S/A provided financial support in the form of author' salary to MADP but did not have any additional role in the study design, data collection and analysis, decision to publish, or preparation of the manuscript.

The authors Fábio Leivas Leite and Victor Fernando Büttow Roll were supported by grants from Conselho Nacional de Desenvolvimento Científico e Tecnológico, Brazil (CNPq/Produtividade em Pesquisa). The Author Aline Arassiana Piccini Roll was supported by grant from CAPES - Coordenação de Aperfeiçoamento de Pessoal de Nível Superior, Brazil. The funders had no role in study design, data collection and analysis, decision to publish, or preparation of the manuscript.

Competing interests

The MADP received salary from Brasil Foods S/A. This does not alter our adherence to PLOS ONE policies on sharing data and materials. 

The other authors declare no potential conflict of interest.

5. Review Comments to the Author

Reviewer #1: In general, the manuscript clearly presents a straightforward meta-analysis, although the number of studies entering the meta-analysis is somewhat on the low side. There are some very occasional issues with English language idiomatic usage, however, overall the writing is quite acceptable.

The following are generally minor comments:

1) Line 147: It is a bit unclear what the authors mean by "there were no repetitions of publications". Does this mean that there was only a single publication covering this treatment?

Thanks reviewer for the valuable comments provided. The sentence was adjusted as follows: It was identified that for some treatments and response variables there was only a single published study.

2) From the PRISMA checklist: "Describe all information sources (e.g., databases with dates of coverage, contact with study authors to identify additional studies) in the search and date last searched."

Did the authors contact study authors?

We have updated our search strategy in Materials and Methods section as follows: Although the search was extensive, authors were not contacted to ascertain further information or to obtain unpublished work. However, the possibility of publication bias due to the existence of unpublished studies was evaluated using the funnel plot technique and the Egger's test.

3) From the PRISMA checklist: "Describe methods used for assessing risk of bias of individual studies (including specification of whether this was done at the study or outcome level), and how this information is to be used in any data synthesis."

This was not performed.

The inclusion criteria defined in the manuscript naturally excluded those poorly designed studies. For this reason, we did not assess the risk of bias of individual studies.

On the other hand, following the recommendations of the PRISMA method, we included the analysis of publication bias that we believe provides more relevant information about the validity of the results found in the manuscript. 

Reviewer #2: 1- The authors rigorously explain their research criteria and the conclusions are drawn appropriately based on the data presented.

2- The data criteria and data analyses were thoroughly explained and support the conclusions.

3- The authors have provided supplemental materials that that support their findings.

4- The manuscript has minor grammatical errors that need to be fixed, but it is written in an orderly fashion.

Thanks reviewer for the valuable comments provided.

Sincerely,

Victor Fernando Roll, Prof. Dr.

Department of Animal Science

Federal University of Pelotas

Roll98@ufpel.edu.br

---

## [Editor Report · Decision Letter 1]

23 Apr 2020

An assessment of the impacts of litter treatments on the litter quality and broiler performance: a systematic review and meta-analysis

PONE-D-20-02805R1

Dear Dr. Roll,

We are pleased to inform you that your manuscript has been judged scientifically suitable for publication and will be formally accepted for publication once it complies with all outstanding technical requirements.

With kind regards,

Arda Yildirim, Ph.D.

Academic Editor

PLOS ONE

Additional Editor Comments (optional):

Thank you for responding to all comments and for revising the manuscript. Best wishes,
---

## [Editor Report · Acceptance letter]

27 Apr 2020

PONE-D-20-02805R1 

An assessment of the impacts of litter treatments on the litter quality and broiler performance: a systematic review and meta-analysis 

Dear Dr. Roll:

I am pleased to inform you that your manuscript has been deemed suitable for publication in PLOS ONE. Congratulations! Your manuscript is now with our production department. 

With kind regards,

on behalf of

Dr. Arda Yildirim 

Academic Editor

PLOS ONE